

# The scavenger receptor repertoire in six cnidarian species and its putative role in cnidarian-dinoflagellate symbiosis

Emilie F. Neubauer[1], Angela Z. Poole[2], Olivier Detournay[3,4], Virginia M. Weis[3] and Simon K. Davy[1]

[1] School of Biological Sciences, Victoria University of Wellington, Wellington, New Zealand
[2] Department of Biology, Western Oregon University, Monmouth, OR, United States
[3] Department of Integrative Biology, Oregon State University, Corvallis, OR, United States
[4] PLANKTOVIE sas, Allauch, France

Corresponding author
Virginia M. Weis,
weisv@oregonstate.edu

## ABSTRACT

Many cnidarians engage in a mutualism with endosymbiotic photosynthetic dinoflagellates that forms the basis of the coral reef ecosystem. Interpartner interaction and regulation includes involvement of the host innate immune system. Basal metazoans, including cnidarians have diverse and complex innate immune repertoires that are just beginning to be described. Scavenger receptors (SR) are a diverse superfamily of innate immunity genes that recognize a broad array of microbial ligands and participate in phagocytosis of invading microbes. The superfamily includes subclades named SR-A through SR-I that are categorized based on the arrangement of sequence domains including the scavenger receptor cysteine rich (SRCR), the C-type lectin (CTLD) and the CD36 domains. Previous functional and gene expression studies on cnidarian-dinoflagellate symbiosis have implicated SR-like proteins in interpartner communication and regulation. In this study, we characterized the SR repertoire from a combination of genomic and transcriptomic resources from six cnidarian species in the Class Anthozoa. We combined these bioinformatic analyses with functional experiments using the SR inhibitor fucoidan to explore a role for SRs in cnidarian symbiosis and immunity. Bioinformatic searches revealed a large diversity of SR-like genes that resembled SR-As, SR-Bs, SR-Es and SR-Is. SRCRs, CTLDs and CD36 domains were identified in multiple sequences in combinations that were highly homologous to vertebrate SRs as well as in proteins with novel domain combinations. Phylogenetic analyses of CD36 domains of the SR-B-like sequences from a diversity of metazoans grouped cnidarian with bilaterian sequences separate from other basal metazoans. All cnidarian sequences grouped together with moderate support in a subclade separately from bilaterian sequences. Functional experiments were carried out on the sea anemone *Aiptasia pallida* that engages in a symbiosis with *Symbiodinium minutum* (clade B1). Experimental blocking of the SR ligand binding site with the inhibitor fucoidan reduced the ability of *S. minutum* to colonize *A. pallida* suggesting that host SRs play a role in host-symbiont recognition. In addition, incubation of symbiotic anemones with fucoidan elicited an immune response, indicating that host SRs function in immune modulation that results in host tolerance of the symbionts.

## INTRODUCTION

Cnidarians such as reef-building corals engage in an intimate mutualistic symbiosis with photosynthetic dinoflagellates in the genus *Symbiodinium* that together form the trophic and structural foundation of coral reef ecosystems. *Symbiodinium* spp. provide large amounts of reduced organic carbon to the host in exchange for inorganic nutrients, a high light environment and refuge from herbivory (*Yellowlees, Rees & Leggat, 2008*). In the majority of cnidarian-*Symbiodinium* interactions, the symbionts are taken up by host cells *via* phagocytosis. Instead of being digested as food, the symbionts resist host destruction and persist in host cells by residing in vacuoles known as symbiosomes (*Davy, Allemand & Weis, 2012*) The molecular interplay between host cnidarian and resident symbionts during both the establishment and ongoing maintenance of the symbiosis is critical for a healthy holobiont (*Weis & Allemand, 2009*).

Animal innate immune systems are central to managing microbes by both tolerating and promoting the survival of beneficial symbionts and resisting and destroying negative invaders (*Bordenstein & Theis, 2015*; *McFall-Ngai et al., 2013*; *Schneider & Ayres, 2008*). With the increased availability of sequence resources, there is now ample evidence that innate immune pathways are ancestral and that basal metazoans including cnidarians possess many of these pathways originally described in mammals and flies (*Fuess et al., 2016*; *Miller et al., 2007*; *Yuen, Bayes & Degnan, 2014*). Furthermore there are numerous examples of expansions of some innate immune gene families in invertebrates that are larger than those in vertebrate genomic repertoires, including NOD-like receptors, scavenger receptors, TIR-domain-containing proteins and ficolins (*Baumgarten et al., 2015*; *Buckley & Rast, 2015*; *Hamada et al., 2013*; *Pancer, 2000*; *Poole & Weis, 2014*; *Shinzato et al., 2011*). A class of well-described host-microbe molecular interactions mediated by innate immunity are the PRR-MAMP interactions where microbe-associated molecular patterns (MAMPs) on the surface of microbes, such as lipopolysaccharide or glycans, are recognized by pattern recognition receptors (PRRs) on the surface of host cells (*Janeway & Medzhitov, 2002*). These steric interactions launch a series of downstream signalling cascades in the host that serve to resist and destroy negative invaders or tolerate and nurture positive microbes. Genomic and transcriptomic studies of cnidarians are revealing the presence of many classical PRRs that have been extensively characterized in higher metazoans (*Fuess et al., 2016*; *Miller et al., 2007*).

One group of PRRs in the Metazoa are the scavenger receptors (SRs), so-named for their role in the scavenging and clearing of microbial invaders, modified host molecules, and apoptotic cell debris (*Areschoug & Gordon, 2009*; *Canton, Neculai & Grinstein, 2013*). SRs have a high affinity for a wide range of ligands and this flexibility of ligand binding has led them to be described as 'molecular fly paper' (*Krieger, 1992*). A key role of SRs in innate immune function is their action as PRRs on phagocytic cells where they mediate direct non-opsonic phagocytosis of pathogenic microbes (*Areschoug & Gordon, 2009*) SRs are thought to engage in heteromultimeric signalling complexes, known as signalosomes, involving multiple PRRs and other molecules that together effect signal transduction in cells, thereby alerting them to microbes or modified host molecules

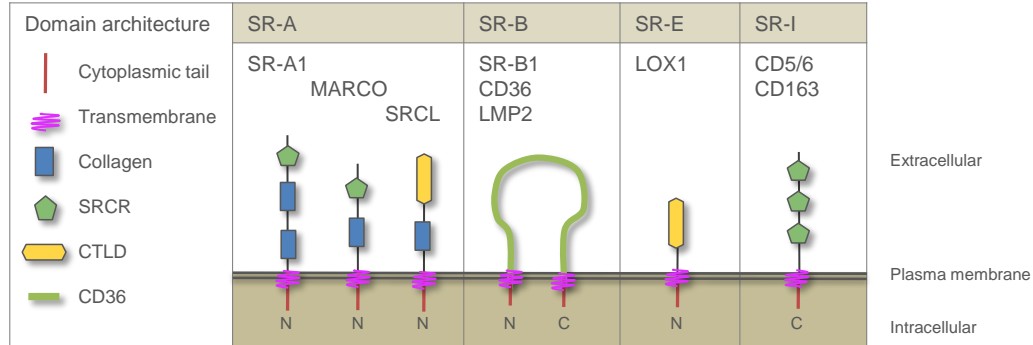

**Figure 1  Domain architecture of vertebrate SRs relevant to this study.** All SR sequences are anchored in the membrane with one or two transmembrane domains. All have very short cytoplasmic tails and extensive extracellular ligand-binding domains. SR-As contain a collagen domain(s) and can include an SRCR or a CTLD. SR-Bs have two cytoplasmic tails on either side of a CD36 domain that forms an extracellular loop. SR-Es are defined by the presence of a CTLD. SR-Is have multiple SRCR repeats and no other identifiable extracellular domains. C, carboxy terminus; CTLD, C type lectin domain; LOX1, lectin-like oxidized low density lipoprotein receptor 1; MARCO, macrophage receptor with collagenous structure; N, amino terminus; SRCL, scavenger receptor with C-type lectin; SRCR, scavenger receptor cysteine-rich domain.

(*Canton, Neculai & Grinstein, 2013*). The SR superfamily is a large group of structurally diverse transmembrane cell surface glycoproteins, divided into nine classes SR-A through SR-I (*Canton, Neculai & Grinstein, 2013*; *Krieger, 2001*). The classes have overlapping specificities that result in an enormous breadth of MAMP recognition (*Krieger, 1992*). Members within a given class share some sequence homology, with little-to-no homology occurring between classes. The classes are grouped by their multiple domains with no single domain common to all (*Gordon, 2002*; *Gough & Gordon, 2000*). SR domains occur on the extracellular portion of the protein; the proteins are anchored in the cell membrane with transmembrane domain(s) and contain short cytoplasmic tail(s). Figure 1 depicts the four SR classes that are relevant to this study. SRs are a potential target for manipulation by invading parasites, pathogens and potentially mutualists. Several pathogens have evolved mechanisms to evade SR-mediated recognition (*Areschoug & Waldemarsson, 2008*; *Faure & Rabourdin-Combe, 2011*). Indeed, certain human pathogens exploit specific SRs for their own benefit. For example, the Hepatitis C virus (HCV) (*Catanese et al., 2007*) and the malaria parasite *Plasmodium falciparum* (*Ndungu et al., 2005*; *Rodrigues et al., 2008*) have surface ligands that are recognized by SR-B1, and both use this recognition to gain entry to host cells.

SR-As and SR-Is contain the scavenger receptor cysteine rich (SRCR) domain, which consists of a 110 aa motif with conserved spacing of six to eight cysteines (*Hohenester, Sasaki & Timpl, 1999*). The SRCR domain is found in a wide range of membrane and soluble proteins and often occurs in multiple repeats arrayed on the protein (*Hohenester, Sasaki & Timpl, 1999*; *Martinez et al., 2011*; *Sarrias, Grønlund & Padilla, 2004*). Some SR-As and SR-Es contain C-type lectin domains (CTLDs), a common domain in many proteins, that are often involved in lectin-glycan interactions (*Cambi, Koopman & Figdor, 2005*). SR-Bs contain the CD36 domain and have two cytoplasmic tails rooted in the membrane

with two transmembrane regions, forming an extracellular loop (*Silverstein & Febbraio, 2009*). SR genes encoding SRCR, CTLD and CD36 domains have been described in invertebrates (*Hibino et al., 2006*; *Lehnert et al., 2014*; *Pancer et al., 1997*; *Schwarz et al., 2007*; *Wood-Charlson & Weis, 2009*). However a detailed bioinformatic characterization of cnidarian SR genes homologous to vertebrate SR-As, SR-Bs, SR-Es and SR-Is is lacking as are any studies exploring the function of these proteins.

SRs are of interest in studies of cnidarian immunity and symbiosis. First, interactions between SR-E-like host lectin-like proteins and symbiont surface glycans play an important role in host-symbiont recognition during onset of symbiosis (reviewed in *Davy, Allemand & Weis, 2012*). In addition, SR-B homologues in two species of sea anemone, *Anthopleura elegantissima* (*Rodriguez-Lanetty, Phillips & Weis, 2006*) and *Aiptasia pallida* (*Lehnert et al., 2014*) were found to be highly expressed in symbiotic compared to aposymbiotic individuals. For *A. pallida* this was a dramatic difference in expression where symbiotic anemones had 28-fold greater expression than aposymbiotic animals. These studies suggest that SR-E and SR-B homologues are playing a role in host-symbiont communication.

There were two aims for this study. The first was to identify SRs in six cnidarian species, all in Class Anthozoa (corals, sea anemones and others), using a variety of genomic and transcriptomic resources, and compare the repertoire to vertebrate SRs of known function. This provides a platform for identifying potential roles of cnidarian SR proteins in immunity and symbiosis. The second aim was to perform simple functional experiments to examine the role of SRs in symbiont recognition and uptake by the sea anemone *A. pallida*, a well-studied model system for the study of coral-dinoflagellate symbiosis. We hypothesized that if a symbiont is co-opting host SRs to initiate tolerogenic pathways that dampen or prevent an immune response, blocking SR-ligand-binding capabilities would induce an immune response.

## MATERIALS AND METHODS

### Anthozoan genomic and transcriptomic resources

To characterize the SR protein repertoire in cnidarians, six species with publically available resources were searched. These included three anemone species: *Anthopleura elegantissima* (*Kitchen et al., 2015*), *Aiptasia pallida* (*Baumgarten et al., 2015*; *Lehnert, Burriesci & Pringle, 2012*), and *Nematostella vectensis* (*Putnam et al., 2007*), and three coral species: *Acropora digitifera* (*Shinzato et al., 2011*), *A. millepora* (*Moya et al., 2012*) and *Fungia scutaria* (*Kitchen et al., 2015*). These species were selected based on the availability of transcriptomic and genomic resources and to include a diversity of organisms within Class Anthozoa. These resources were derived from various developmental stages and symbiotic states (Table 1). All resources were used as provided, with the exception of the *A. pallida* transcriptome, for which raw Illumina sequence reads for accession number SRR696721 were downloaded from the sequence read archive entry (http://www.ncbi.nlm.nih.gov/sra/SRX231866) and reassembled using Trinity (*Grabherr et al., 2011*) resulting in a better assembly than the original one performed.

**Table 1** **Information on the cnidarian sequence resources used in this study.** Non-symbiotic refers to species that do not form symbioses with dinoflagellates. Aposymbiotic refers to species that do form symbioses but the material from which the sequencing was performed did not contain symbionts.

| Organism | Developmental stage | Symbiotic state | Data type | Reference |
|---|---|---|---|---|
| *Nematostella vectensis* | Larvae | Non-symbiotic | Genome | *Putnam et al. (2007)* |
| *Anthopleura elegantissima* | Adult | Aposymbiotic | Transcriptome | *Kitchen et al. (2015)* |
| *Aiptasia pallida* | Adult | Aposymbiotic | Transcriptome | *Lehnert, Burriesci & Pringle (2012)* |
| *Aiptasia pallida* | Adult | Aposymbiotic | Genome | *Baumgarten et al. (2015)* |
| *Acropora digitifera* | Sperm | Aposymbiotic | Genome | *Shinzato et al. (2011)* |
| *Acropora millepora* | Adult and larvae | Symbiotic | Transcriptome | *Moya et al. (2012)* |
| *Fungia scutaria* | Larvae | Aposymbiotic | Transcriptome | *Kitchen et al. (2015)* |

## SR sequence searching

Twenty-four non-cnidarian sequences were obtained, primarily from GenBank and other publically available databases (Table S1), for use in creating multiple sequence alignments and phylogenetic trees. Eleven human SR genes were chosen for production of reference protein domain architecture diagrams, to compare predicted cnidarian proteins with human SR proteins of known function (Fig. 2).

To search for cnidarian SR proteins, databases were queried using several search strategies to ensure all sequences were recovered. BLASTp or tBLASTn searches with mouse and human SR protein sequences (SR-A1, MARCO, SRCL, CD36, SRB1, LMP2, and LOX1) (File S1) and consensus sequences (pfam01130: CD36, pfam00530: SRCR) from the conserved domain database (http://www.ncbi.nlm.nih.gov/cdd) as queries were performed for each resource. Keyword searches were used with the terms SR, CD36, LMP2, SRCR, and scavenger where GO or KEGG annotations were available. Lastly, representative *N. vectensis* sequences of each protein type (SRCR-domain-containing, CD36, SRB1, and LOX1) were also used as queries for tBLASTn searches of the other five cnidarian resources. A high $e$-value cutoff ($1 \times 10^{-1}$) was used in the BLAST searches to recover divergent sequences. All BLAST searches were performed using the default settings in Geneious pro version 7.1.8 with the exception of *N. vectensis*, for which searches were performed through the Joint Genome Institute online portal using the default settings (*Kearse et al., 2012*). A list of metazoan resources searched are listed in Table S1. Blast query sequences and cnidarian sequences identified are tabulated in File S1.

To confirm that the sequences obtained contained SR domains, nucleotide sequences were translated using Geneious and then annotated using the InterProScan plugin (*Quevillon et al., 2005*). Only sequences in which two or more databases within InterProScan found either SRCR, CD36, or CTLD domains with an $e$-value of $<1 \times 10^{-4}$ were used. Where the InterProScan plugin was unable to resolve protein domains, (this occurred for approximately 1 in 10 sequences) the sequences were analysed using the online protein domain database PfamA (http://pfam.sanger.ac.uk) using the default program settings. Sequences for each species were aligned and those that were identical or almost identical (<5 aa difference in the conserved domains) were omitted from the

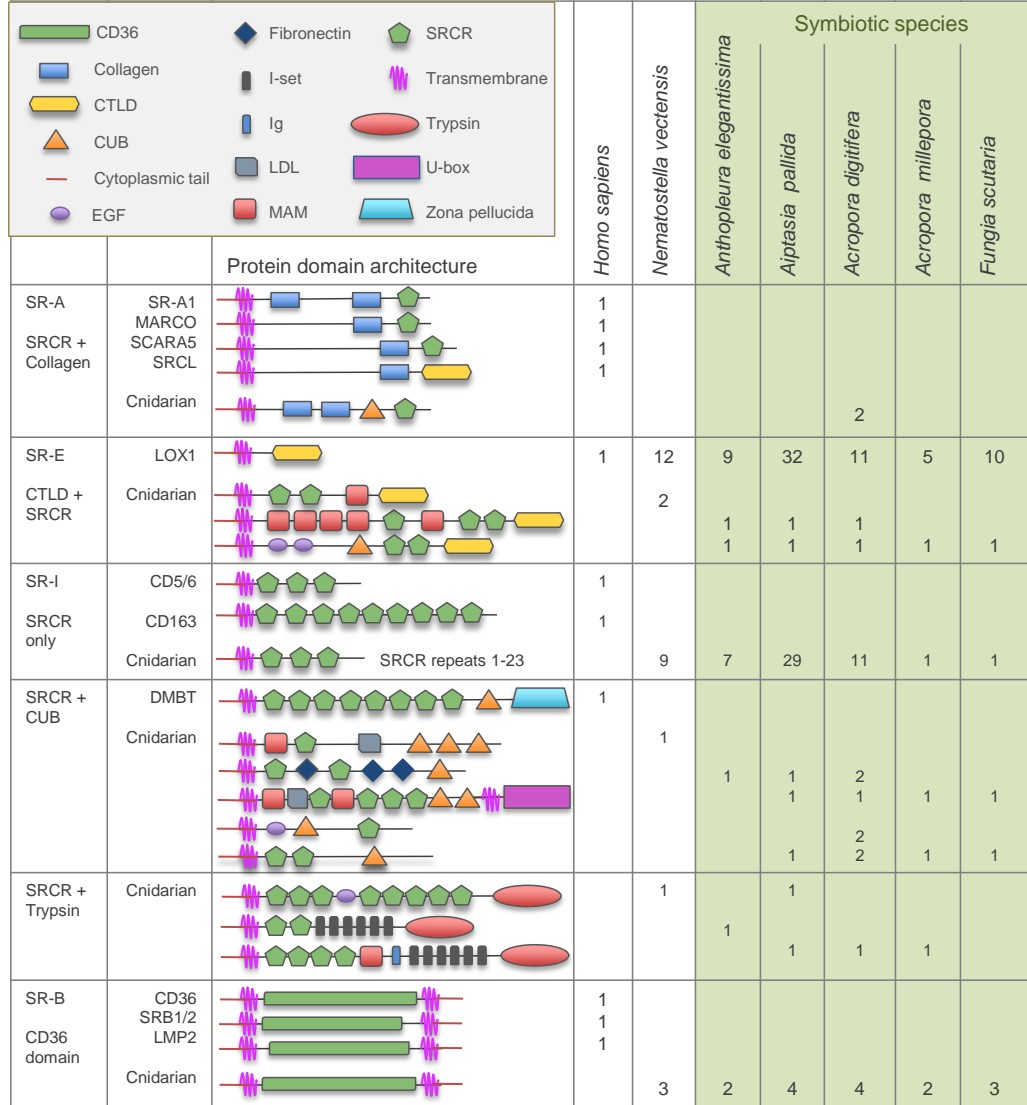

**Figure 2  Domain architecture of cnidarian SR domains in the six resources searched compared to human SRs.** Identified cnidarian SR-A-like and SR-E-like sequences display diverse domain architecture and include novel domain combinations not found in vertebrates. SR-I-like sequences had a varying number of SRCR repeats. A variety of SRCR-domain-containing cnidarian sequences identified did not fit the criteria of any vertebrate SR classes and are presented as SRCR + CUB domains or SRCR + trypsin domains. SR-B-like domain combinations closely resembled vertebrate SR-Bs with two transmembrane domains, two cytoplasmic tails and a CD36 domain. CTLD, C type lectin domain; CUB, complement C1r/C1s, Uegf, BMP1; DMBT, deleted in malignant brain tumor protein; Ig, immunoglobulin; I-Set, intermediate set of immunoglobulin domain; LDL, low density lipoprotein; LOX1, lectin-like oxidized low density lipoprotein receptor 1; EGF, epidermal growth factor; MAM meprin/A5-protein/PTPmu; MARCO, macrophage receptor with collagenous structure; SCARA5, scavenger receptor class A member 5; SRCL, scavenger receptor with C-type lectin; SRCR, scavenger receptor cysteine-rich domain; U-box, ubiquitin box. Human SR data taken from *Canton, Neculai & Grinstein (2013)* (See File S1 for sequence information.).

analysis as they likely represented artifacts of assembly or different isoforms of the same protein. Sequences missing a start or stop codon were removed from the analysis.

Only proteins that showed significant PfamA matches to a CTLD, SRCR and/or CD36 domains were included in the study. Diagrammatic representations of the protein domain configurations were produced using this information. Protein domain architectures were grouped together according to common domains and compared to known human SR proteins (Fig. 2).

## Phylogenetic analysis of SR-B homologues

A multiple sequence alignment of a subset of CD36-domain-containing sequences was performed with the MAFFT v 7.017 plug-in (*Katoh et al., 2002*) through Geneious (*Kearse et al., 2012*), using the default settings. The program ProtTest v2.4 (*Abascal, Zardoya & Posada, 2005*) was used to apply AIC1, AIC2 and BIC2 model selection criteria to a variety of possible substitution matrices and rate assumptions to obtain the best-fit model of protein evolution. The results from the overall comparison of these metrics indicated that the best-fit model for the full-length alignment was WAG+G+F. A maximum likelihood tree was produced using FastTree v2.1.5 (*Price, Dehal & Arkin, 2010*). Bootstrap support values were generated using the online program SEQBOOT (*Felsenstein, 2005*) and values above 0.6 support were displayed at the nodes. A PhyML (*Guindon et al., 2005*) alternate tree produced identical topology (data not shown).

## Maintenance and preparation of anemone and dinoflagellate cultures

Symbiotic *A. pallida* cultures were maintained in saltwater aquaria at 26 °C with a 12/12 h light/dark photoperiod, and were fed twice weekly with live brine shrimp nauplii. Animals were rendered aposymbiotic by incubation for 8 h at 4 °C twice weekly for six weeks, followed by maintenance in the dark for approximately one month. Anemones were fed twice weekly with brine shrimp and cleaned of expelled symbionts and food debris regularly.

Cultured dinoflagellates, *Symbiodinium minutum*, clade B1 (culture ID: CCMP830) were maintained in 50 ml flasks in sterile Guillard's f/2 enriched seawater culture medium (Sigma, St. Louis, MO, USA). Dinoflagellate cultures were maintained at 26 °C on a 12/12 h light/dark photoperiod.

In preparation for experimental manipulations, individual anemones were placed in 24-well plates in 2.5 ml of 1-$\mu$m filtered seawater (FSW) and acclimated to the well-plate for 3–4 days, with the FSW replaced daily. Well-plates containing aposymbiotic anemones were kept in the dark and symbiotic anemones were maintained in an incubator at 26 °C with a 12/12 h light/dark photoperiod. Animals were not fed during the experimental time period.

## Addition of fucoidan to block SR binding function

To explore a role for SRs in the onset of symbiosis, fucoidan, a known SR ligand, was added to anemones to block SR binding sites. Fucoidan is a protein derived from the brown alga *Fucus vesiculosus*; this polyanionic ligand is known to bind SRCR and CD36 domains in SR-As and SR-Bs respectively (*Dinguirard & Yoshino, 2006*; *Hsu et al., 2001*; *Thelen et al., 2010*).

To examine the effect of blocking SR ligand binding capabilities on symbiont colonization success, aposymbiotic anemones ($n = 3$ per treatment per time point) were pre-incubated in fucoidan (Sigma, St. Louis, MO, USA), at a concentration of 0 (FSW control), 100, 200 and 400 µg/ml for 18 h, according to Bowdish Lab protocols (online at McMaster University; www.bowdish.ca/lab/protocols). Fucoidan-treated aposymbiotic anemones were subsequently re-inoculated with *S. minutum* CCMP830. CCMP830 cells were pelleted from the culture medium, re-suspended in FSW, and then added to anemones in wellplates to a final concentration of $2 \times 10^5$ symbionts per ml. After incubation for 12 h at 26 °C in the light, anemones were rinsed twice with FSW and fucoidan treatments were refreshed. To test the effect of fucoidan exposure on host health, a second control treatment (fucoidan-washed control) was prepared where aposymbiotic anemones were pre-incubated in 200 µg/ml fucoidan for 18 h, and then washed with FSW prior to being inoculated with symbionts as described above. Anemones for all treatments were sampled at 48 and 96 h postinfection (three tentacles per anemone, for $n = 3$ anemones per treatment per time point).

A second experiment was designed to explore a role for SR binding in host immune tolerance during symbiosis. We hypothesized that if a symbiont is co-opting host SRs to initiate tolerogenic pathways (such as the TGFβ pathway) that dampen or prevent an immune response, blocking SR-ligand-binding capabilities could induce an immune response upon the addition of lipopolysaccharide (LPS). LPS is a MAMP that has been shown to induce an anemone immune response measured as increased nitric oxide (NO) production (*Detournay et al., 2012*; *Perez & Weis, 2006*). Symbiotic anemones were incubated at increasing concentrations of fucoidan: 0 (FSW control), 100, 200, 400 and 800 µg/ml, for 4 h, prior to the addition of 1 µg/ml of LPS (Sigma, St. Louis, MO, USA) (dissolved in 0.1% v/v DMSO) for a further 12 h. The FSW control was also exposed to 1 µg/ml LPS for 12 h. NO production by hosts was quantified as described below.

## Quantifying colonization success and host NO production using confocal microscopy

Colonization success was assessed fluorometrically by confocal microscopy, following methods described in detail by *Detournay et al. (2012)*. Briefly, following experimental manipulation, solutions in wells containing anemones were replaced with 1 ml of relaxing solution (1:1 0.37 M $MgCl_2$: FSW). Samples were observed under a Zeiss LSM 510 Meta microscope with a 40x/0.8 water objective lens and a working distance of 0.8–3.2 mm. Before image scanning, the focal plane of the optical section was adjusted to include the gastrodermal cells within the anemone tentacle. For each experiment, all images were obtained with the same software scanning settings, including detector gain and laser intensity. *S. minutum* cells present were visualized by detecting chlorophyll autofluorescence with excitation and emission wavelengths of 543 and 600–700 nm, respectively. Fluorescence was quantified by first defining the gastrodermal tissue area within the anemone tentacles as a region of interest and then measuring the mean fluorescence intensity (MFI) for that region with the LSM 5 software. Intensity of chlorophyll autofluorescence for each pixel was measured and a threshold value

corresponding to the background was defined by measuring the MFI at 600 nm of a gastrodermal region without symbionts (threshold MFI = 20). Colonization success was expressed as percent of pixels with autofluorescence intensity above the threshold. In colonization experiments, each treatment represents a sample size of three anemones per treatment and time-point, with percent colonization taken as the mean of six tentacles per anemone. Three untreated symbiotic anemones (six tentacles per anemone) were examined to determine a baseline colonization level for symbiotic anemones.

The statistical significance of colonization success under the treatments described above was assessed using a mixed-effects model. As measures on multiple samples (i.e., tentacles) per anemone violate independence assumptions, a mixed effect was used, treating anemone as a random effect to account for correlation among samples with anemones. Main effects included time and treatment, and their interaction was estimated to account for differences between treatments at each time point. The full model can be written as:

$$Y_{i,j} = \beta X_i + \mu_j + \epsilon_{i,j}.$$

Here, $Y_{i,j}$ is the logarithm of percent colonization (plus a small constant) of tentacle $i$ within anemone $j$, $\beta$ is a vector of effects to be estimated, $X$ is a design matrix encoding the treatment and time point, as well as interaction term contrasts, $\mu_j$ is a normally distributed random effect for anemone $j$, and $\epsilon_{i,j}$ are normally distributed residuals. The model was estimated using the LME4 packages (*Bates et al., 2015*) for the statistical computing software R (*R Development Core-Team, 2012*); the script and data used for statistical analyses are given in Files S2–S4.

To measure and visualize production of NO by confocal microscopy, animals were treated as described in detail previously (*Detournay et al., 2012*; *Detournay & Weis, 2011*). Animals were transferred to a microfuge tube containing 500 µl of relaxing solution and 15 µM 4-amino-5-methylamino-2,7 difluorofluorescein diacetate (DAF-FM DA, Molecular Probes, Eugene, OR, USA) with excitation and emission wavelengths of 488 and 510–530 nm, respectively. Samples were incubated for 30 min in the dark and then rinsed twice with relaxing solution. Fluorescence of the DAF-FM DA probe was quantified as described above for chlorophyll autofluorescence quantification.

The statistical analysis of this experiment used a similar model as above, but treating the fucoidan concentration as a continuous variable and fitting a linear slope to the fluorescence intensity. We also added a random effect for tentacle within anemone to account for non-independence of readings within each tentacle. Using the notation above, the model can be written as

$$Y_{k,i,j} = \alpha + \beta F_k + \mu_j + \gamma_{i(j)} + \epsilon_{k,i,j}.$$

Here, $Y_{k,i,j}$ is the fluorescence reading $k$ within tentacle $i$ of anemone $j$, $\alpha$ is the intercept and $\beta$ is the slope of the regression line relating fluorescence to fucoidan concentration $F_k$, $\mu_j$ is a normally distributed random effect for anemone $j$, $\gamma_{i(j)}$ is a normally distributed random effect for tentacle $i$ within anemone $j$, and $\epsilon_{k,i,j}$ are normally distributed residuals.

## RESULTS

Annotated predicted cnidarian SR proteins are illustrated according to their domain architecture and compared with known human SR protein domain organization (Fig. 2). Overall, cnidarian SR-like proteins fall into four groups: SR-As, SR-Es, SR-Is and SR-Bs. The SRCR domain is present in all groups except the SR-Bs.

### Cnidarian SRCR-containing proteins

Vertebrate SR-As are defined by a collagen domain coupled with most proteins containing either an SRCR domain or a CTLD at the C terminus (*Bowdish & Gordon, 2009*). Only two sequences meeting these criteria were identified in the cnidarian resources searched. Both are in *A. digitifera* and contain a CUB domain in addition to two collagen domains and one SRCR. Human SR-Es are defined by the presence of only CTLDs (*Zani et al., 2015*). The human lectin-like oxidized low-density lipoprotein receptor 1 (LOX1) has an N-terminal cytoplasmic tail, a transmembrane domain and a single C-terminal CTLD (*Canton, Neculai & Grinstein, 2013*). Numerous LOX1-like sequences were identified in all of the cnidarian resources searched. SR-Is in humans are defined by containing only SRCR domains in various numbers of repeats and are grouped into three classes: CD5, CD6 and CD163. SR-I-like sequences are abundant in all cnidarian resources, in the same configurations as human SR-Is. SRCR repeat numbers range from one to twenty-three.

A variety of SRCR-domain-containing proteins were also identified in cnidarian sequence resources that could not be classified into any of the vertebrate classes of scavenger receptors. Several cnidarians genes with SRCRs and CUB domains were identified that resemble 'human deleted in malignant brain tumor' (DMBT) protein that contains eight SRCR repeats, a single CUB domain and a zona pelucida domain at the C-terminal end. Predicted cnidarian proteins that resemble DMBT contain one to three CUB domains combined with a range of other protein domains, including MAM, fibronectin, UBOX and multiple SRCRs. Five of the six cnidarian resources contain sequences with a potentially novel domain configuration of multiple SRCRs, several other domains, including multiple immunoglobulin domains, and a C-terminal trypsin domain.

### Cnidarian SR-B-like proteins

Searches identified eighteen full-length putative cnidarian SR-B sequences, all containing a CD36 domain. Full-length proteins were defined as those containing both transmembrane regions that form the SR-B extracellular loop configuration. Humans have four distinct SR-Bs - CD36, SRB1 & 2, and LMP2 - while the six cnidarian species searched contained between two and four full-length proteins.

### Phylogenetic analysis of SR-B-like proteins

Phylogenetic analysis was carried out on the CD36 domains from SR-B-like sequences identified (Fig. S1 and Fig. 3). Protein sequence alignments of the predicted SR-B-like proteins from cnidarians, combined with a subset of vertebrate and invertebrate sequences, revealed that there is some conservation of the CD36 domain across metazoans. Cnidarian sequences showed weak homology to human SR-Bs, with 26–32%, 28–37% and 28–33%

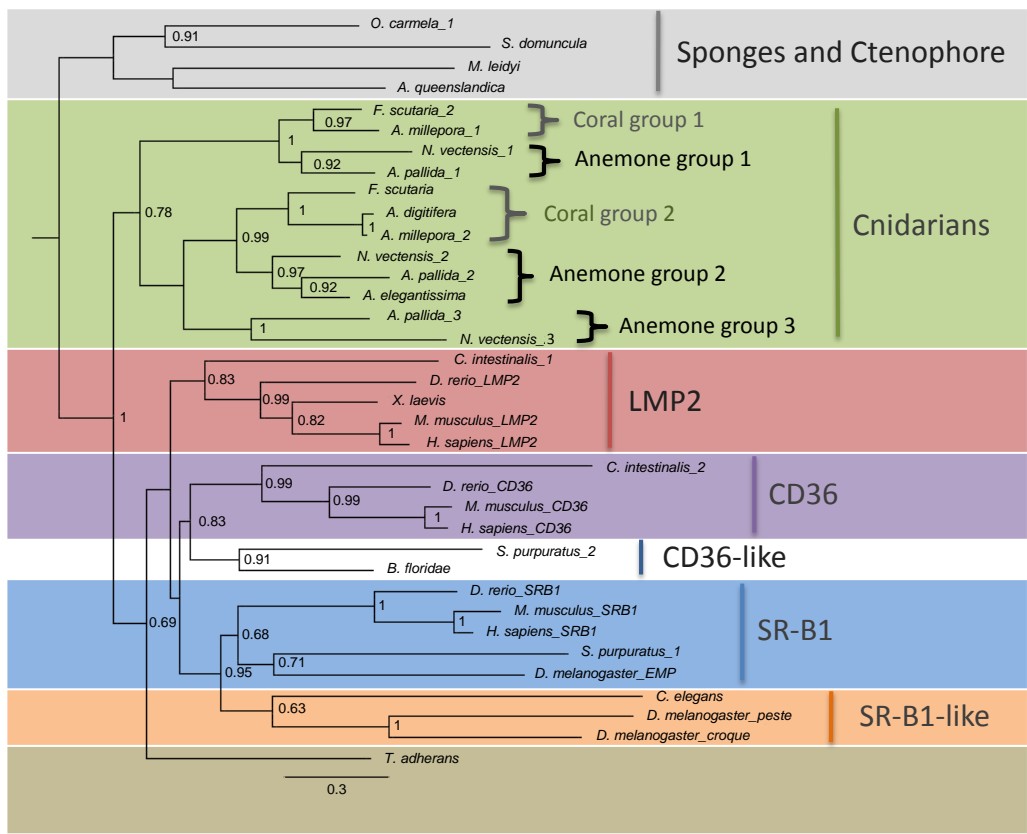

**Figure 3** **Maximum-likelihood tree of SR-Bs from across the Metazoa.** The tree was constructed with the CD36 domain of each protein using FastTree v 2.1.5. Bootstrap support values were generated using SEQBOOT, values above 0.6 are displayed at nodes. The alignment, including organism names, is displayed in Fig. S1.

identity to human CD36, LMP2, and SR-B1, respectively. Identities within the cnidarian group were higher, ranging from 39 to 95% with the two *Acropora* species showing the highest homology to each other. Cnidarian sequences showed between 21 and 27% identity to the predicted SR-B-like protein sequence from the sponge, *Suberites domuncula*. Predicted cnidarian proteins lacked one of the three pairs of cysteine residues known to form three disulfide bridges in the human CD36 protein (Fig. S1) (*Silverstein & Febbraio, 2009*). However, a pair of cysteine residues was found in all cnidarian study species at positions C107 and C117. Predicted cnidarian proteins had eight to ten N-linked glycosylation sites compared with eleven and eight sites in human SR-B1 and CD36 respectively.

Putative cnidarian SRB proteins grouped with high support in a large clade with the bilaterians and separately from other basal metazoans. Within this large clade, cnidarians grouped together, forming a separate clade from the bilaterians. Within the cnidarian clade, there were three well-supported sub-clades, two containing both coral and anemone species and a third, containing only anemone sequences (Fig. 3). Corals and sea anemones sequences formed distinct groupings within each of these clades. In contrast, bilaterian

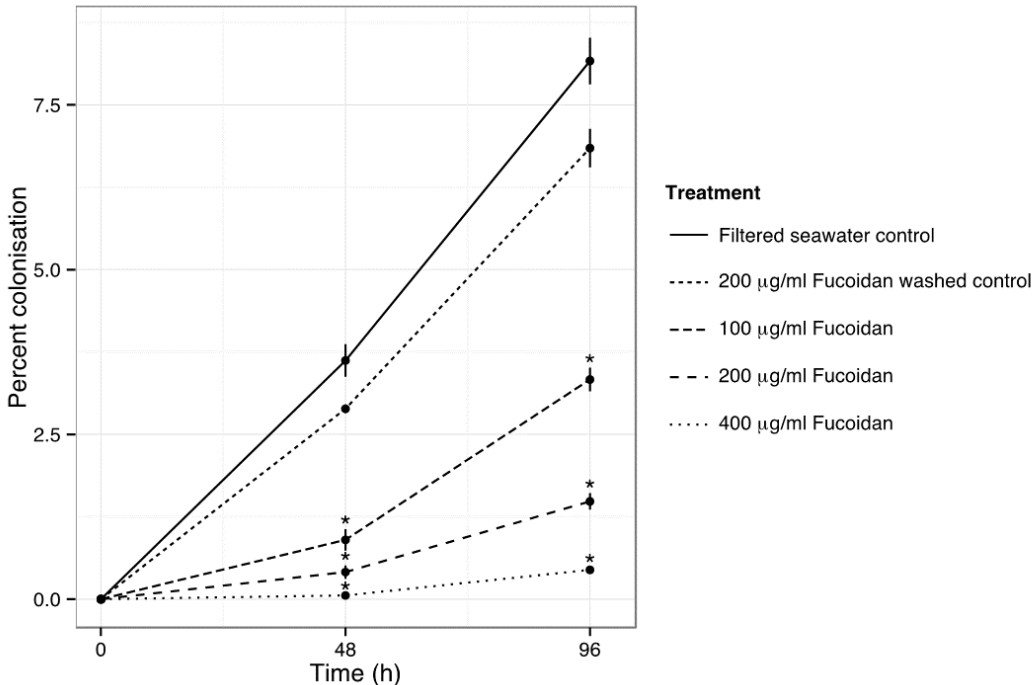

**Figure 4  Experimental colonization by *S. minutum* CCMP830 of aposymbiotic *A. pallida* treated with increasing levels of the SR inhibitor, fucoidan.** Graph shows percent colonization success as measured by surface area of host gastrodermis occupied by symbionts (see Methods for details) as a function of time after inoculation. Two controls were included: FSW alone and an 18 h incubation in 200 $\mu$g/ml fucoidan in FSW followed by a 48 h recovery in FSW to test for fucoidan toxicity to the animals. Anemones in experimental fucoidan treatments exhibited a dose-dependent response with decreased colonization success with increasing fucoidan concentrations. Bars represent means $\pm$ SD, $n = 3$ anemones per treatment. Asterisks indicate high ($p > 0.999$) posterior probability of treatment effects being different from controls under the Bayesian ANOVA model.

invertebrate sequences grouped with mammalian sequences in several different sub-clades of SR-Bs: LMP2, CD36, CD36-like, SRB1, and SR-B1-like proteins.

## Experimental blocking of SR proteins with fucoidan reduces colonization success and elicits an immune response in *A. pallida*

Fucoidan-treated anemones showed significantly lower levels of colonization (0–3%) than either the FSW control or anemones pre-incubated in fucoidan and then rinsed 48 h prior to time zero. Colonization success decreased significantly in a dose-dependent manner (Fig. 4, Bayesian $P < 0.0001$).

A second fucoidan experiment investigated the possible immune-regulation role of an SR in symbiosis maintenance. Symbiotic anemones were treated with increasing concentrations of fucoidan and were subsequently immune-challenged by incubation with LPS. The FSW control-treated anemones had low levels of NO production, a proxy for an immune response, measured as MFI of the NO-specific probe DAF-FM DA in tentacles, in response to incubation in LPS. In contrast, fucoidan-treated anemones showed a significant (Bayesian $P < 0.0001$) dose-dependent response of increasing NO production with increasing concentrations of fucoidan (Fig. 5).

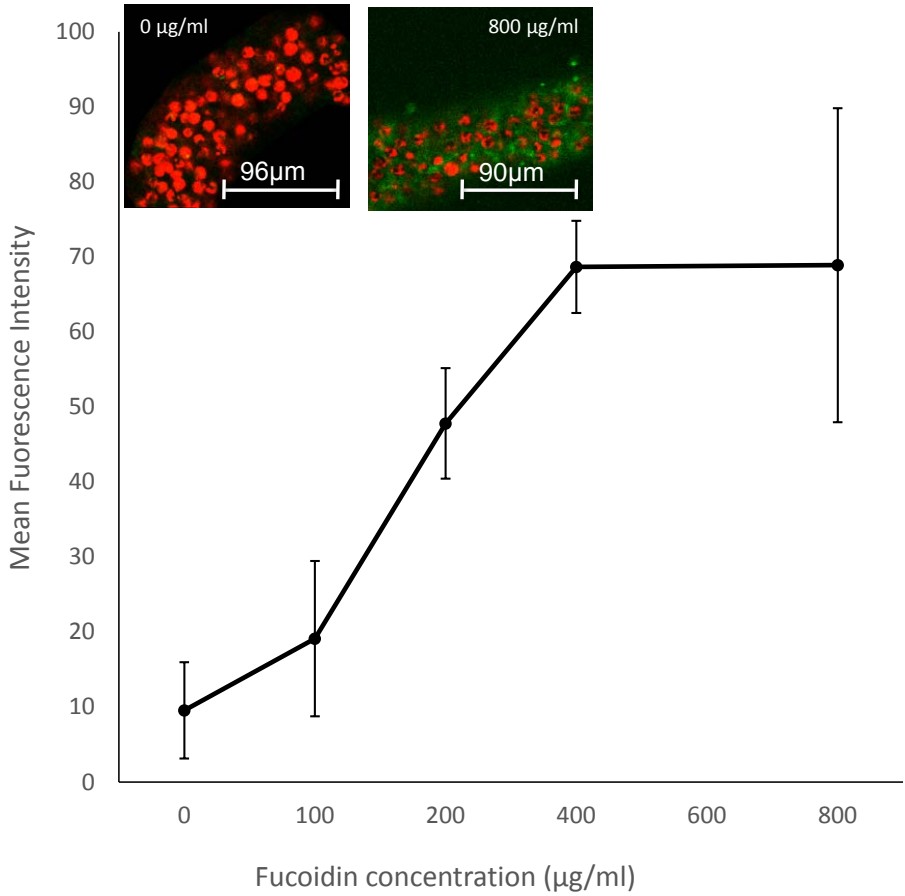

**Figure 5** **Effect of SR inhibition by fucoidan on immune stimulation in symbiotic *A. pallida*.** Immune stimulation of animals was elicited by incubation in 1 μg/ml LPS overnight prior to the experiment. Immune stimulation was measured by quantifying DAF-FM DA, a probe for the presence NO, itself a marker for immune stress. Graph shows MFI of DAF-FM DA in tentacles in response to incubation in increasing concentrations of fucoidan. Animals exhibited a significant dose-dependent response to fucoidan (Bayesian generalized linear mixed model, $P < 0.0001$), with increasing NO production with increasing SR inhibition by fucoidan. Bars represent means ± SD; $n = 3$ anemones. Inset: representative confocal images of tentacles incubated in FSW only and 800 μg/ml fucoidan. DAF-FM DA (green) symbiont autofluorescence (red).

## DISCUSSION

### An expanded SRCR-domain-containing protein repertoire in cnidarians

The SRCR-domain-containing protein repertoire in cnidarians, is expanded compared to that in humans, with the *A. pallida* genome containing the highest number at 36 genes (Fig. 2). This finding is consistent with numerous other studies describing expansions of innate immune gene families in invertebrates (see 'Introduction'). Other examples of SRCR-domain-containing protein repertoire expansion have been described in invertebrates, specifically in the sea urchin, *Strongylocentrotus purpuratus* and the cephalochordate, *Branchiostoma floridae*, which have 218 and 270 SRCR-containing sequences respectively (*Huang et al., 2008*; *Pancer, 2000*; *Pancer, Rast & Davidson, 1999*;

*Rast & Messier-Solek, 2008*). These numbers are high compared to the 16 genes present in humans. In addition, cnidarian SRCR-domain-containing proteins include a variety of genes with novel domain combinations that have not been found in other organisms (Fig. 2). Identification of these novel domain combinations in cnidarian immune gene repertoires is consistent with other studies of basal metazoan immune genes (*Hamada et al., 2013*; *Poole & Weis, 2014*; *Ryu et al., 2016*) The searches for SR genes in the three transcriptomes (Table 1) likely revealed underestimates of the total SR repertoire, given that transcriptomes represent snapshots of the whole genome.

## CTLD-domain-containing SRs in cnidarians

In contrast to the human genome, which contains a single LOX1 gene, all six cnidarian resources searched contained multiple LOX1-like SR-Es (Fig. 2). These searches add to previous characterizations of lectin-like proteins in cnidarians, including in corals and sea anemones (*Jimbo et al., 2005*; *Jimbo et al., 2000*; *Kvennefors et al., 2010*; *Kvennefors et al., 2008*; *Meyer & Weis, 2012*; *Vidal-Dupiol et al., 2009*; *Wood-Charlson & Weis, 2009*). Human LOX1 has a diversity of signalling functions, including in recognition of microbes *via* host CTLD-microbe glycan binding: a PRR-MAMP interaction (*Canton, Neculai & Grinstein, 2013*). In cnidarians, previous studies have detailed a role for lectin-glycan interactions in the establishment of cnidarian-dinoflagellate symbioses (reviewed in *Davy, Allemand & Weis, 2012*). The identification of multiple LOX1-like proteins and several other CTLD-containing proteins with novel domain combinations across the six species examined further strengthens the hypothesis that host CTLD-symbiont glycan binding plays an important role in host innate immunity and host-symbiont recognition. Cnidarian CTLD-domain-containing proteins described here provide potential target proteins for future experimental investigation of the lectin-glycan interactions.

## CD36-domain-containing SRs in cnidarians

Phylogenetic analysis of metazoan CD36 domains from SR-B homologues showed a well-supported clade of cnidarian sequences (Fig. 3). A large analysis including additional sequences from basal metazoans is required to more definitively reveal deep branching patterns of this gene. The observed differing location of cysteine pairs within the CD36 domain in cnidarian sequences compared to vertebrate ones also occurred in other invertebrates (Fig. S1). As with the cnidarians searched, *C. elegans* contained one differing pair and the three sponges, *Oscarella carmella*, *S. domuncula*, and *Amphimedon queenslandica*, and the ctenophore *Mnemiopsis leidyi* had no sequence pairs in common with vertebrates. These differences may explain why antibodies to human and mouse SR-B1 and CD36 failed to label proteins in *A. pallida* in immunoblot experiments (EF Neubauer, 2010, unpublished data).

## Functional experiments suggest that blocking SRs decreases colonization success and increases the stress response to immune challenge in *A. pallida*

Colonization success in aposymbiotic *A. pallida* challenged with *S. minutum* CCMP830 displayed a dose-dependent response to incubation in the SR inhibitor fucoidan, exhibiting

decreasing colonization success with increasing concentrations of fucoidan (Fig. 4). In vertebrates, fucoidan blocks the positively-charged ligand binding sites on SR-As and SR-Bs and can thereby block phagocytic activity in macrophages (*Dinguirard & Yoshino, 2006*; *Hsu et al., 2001*; *Li et al., 2008*) The observed inhibition of colonization in cnidarians suggests that phagocytosis of symbionts is likewise inhibited and provides evidence that one or multiple SRs with SRCR and/or CD36 domains function in host-symbiont recognition during onset of symbiosis.

Previous transcriptomic studies in *A. elegantissima* and *A. pallida* have found SR-B homologues to be upregulated in symbiotic compared to aposymbiotic anemones, suggesting that they play a role in the symbiosis. Our experiments showing that incubation in fucoidan causes a dose-dependent immune response in symbiotic *A. pallida* (Fig. 5), further implicates a role for SRs in immune tolerance and regulation of symbiosis. In previous work on *A. pallida,* we showed that symbiotic anemones produced significantly less NO in response to an immune challenge with LPS than did aposymbiotic animals, suggesting that symbionts are modulating the host immune response (*Detournay et al., 2012*). The increase in this response in symbiotic anemones incubated in fucoidan suggests that this immune modulation involves an SR ligand-binding domain. Such a response is reminiscent of immune modulation by a variety of invading microbes (*Janeway & Medzhitov, 2002*).

In summary, this study provides the first description of the diversity of SRs in cnidarians. Members include proteins with domain combinations that are highly similar to those in vertebrates as well as those that possess novel combinations. Initial functional experiments using the SR inhibitor fucoidan suggest that SRs play a role in the regulation of cnidarian-dinoflagellate symbioses. Future functional studies on candidate SRs identified in this study can further explore their role in cnidarian immunity and symbiosis.

## ACKNOWLEDGEMENTS

We thank Eli Meyer for the reassembly of the *A. pallida* transcriptome and Philipp Neubauer for assistance with statistical analysis. We wish to acknowledge the Confocal Microscopy Facility at the Center for Genome Research and Biocomputing at Oregon State University

### Funding

This work was partially supported by a grant from the National Science Foundation to VMW (IOB0919073). EFN was supported by a Commonwealth Doctoral Scholarship and a Faculty of Science Strategic Research Grant from Victoria University of Wellington. The funders had no role in study design, data collection and analysis, decision to publish, or preparation of the manuscript.

### Grant Disclosures

The following grant information was disclosed by the authors:
National Science Foundation: IOB0919073.
Victoria University of Wellington.

## Competing Interests

The authors declare there are no competing interests. Olivier Detournay is the cofounder of Planktovie.

## Author Contributions

- Emilie F. Neubauer conceived and designed the experiments, performed the experiments, analyzed the data, wrote the paper, prepared figures and/or tables, reviewed drafts of the paper.
- Angela Z. Poole analyzed the data, wrote the paper, reviewed drafts of the paper.
- Olivier Detournay designed the experiments, and provided an immunological perspective.
- Virginia M. Weis conceived and designed the experiments, contributed reagents/materials/analysis tools, wrote the paper, prepared figures and/or tables, reviewed drafts of the paper.
- Simon K. Davy conceived and designed the experiments, contributed reagents/materials/analysis tools, wrote the paper, reviewed drafts of the paper.

## Data Availability

The raw data has been supplied as a Supplementary File.

## Supplemental Information

Supplemental information for this article can be found online at http://dx.doi.org/10.7717/peerj.2692#supplemental-information.

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
