# Peer review of "The scavenger receptor repertoire in six cnidarian species and its putative role in cnidarian-dinoflagellate symbiosis"

_PeerJ, doi:10.7717/peerj.2692_

## Round 0.1 · original submission · Minor Revisions

· Academic Editor

Minor Revisions

Dear Authors. Two referees have now reviewed your manuscript and their comments are below. I suggest you correct your text accordingly. Sincerely.

Reviewer 1 ·

Basic reporting

Your aim was to describe the diversity of SRs in Cnidaria but you just analysed deeply the SR-B family. Please include the analysis of SR-E and SR-I (the same way done for SR-B) in the Results section, since you found signals of expansion on them too.

The experimental step of the work with fucoidan is fine as a first approach.

Some specific comments:

Line 55 – The bootstrap support value from the ancestral node of cnidarians and the rest of bilaterian was 1. It should be written “high support” instead of “moderate support”.

Line 328 – Please include the information of Figure S1 into Figure 2.

Line 346 – There are 18 CD36 for Cnidaria in Figure 2 but you just used 12 in Figure 3. Where are the other six? The missing CD36 are: one from A. elegantissima, one from A. pallida, three from A. digitifera and one from F. scutaria.

Line 346 – “SR-B-like” instead of “SRB-like”.

Lines 348-367 – Please report the results for SR-E and SR-I too.

Line 388 – It is not well explained. You said 72 SRCR-domain-containing protein but 32 of them contains only CTLD domains and four of them have only CD36. Also, the number 72 refers to all domains found for Aiptasia (including five in the supplementary file). Please correct this sentence.

Experimental design

Lines 156-166 – Why these six species? How many cnidarian species with genome sequenced were available when you assembled the data? You could include the date you downloaded the genomes/transcriptomes here. Also, why you reassembled the data of Aiptasia? Please add the answer in the text.

Line 182 – 1x10-1 is not a high e-value. Delete the high, please.

Lines 199-202 – You used just the CD36 domain to reconstruct the phylogenetic tree (line 205) but the way is written here appears that you used these three domains.

Line 204-213 – Why did you not reconstruct the phylogeny of SR-E and SR-I families since there was expansion on these domains in Cnidaria too? Please repeat this analysis (as well as the alignment on Figure S2) for SR-E and SR-I. Also, make the alignments for the three SRs (SR-B, SR-E and SR-I) available (e.g. Dryad repository).

Lines 204-213 – Please cite the references for the software employed: MAFFT, Geneious, ProtTest, FastTree, SEQBOOT and PhyML.

Lines 212-213 – Instead “identical topography” use “identical topology”.

Validity of the findings

Everything is fine. The addition of SR-E and SR-I analysis will improve the relevance of the paper.

Additional comments

No comments.

·

Basic reporting

The article presents interesting results in the diversity of scavenger receptors (SR) in cnidarians and is generally well written. Results are relevant and properly represented by figures.
Background information in the structure and function of SR are well covered in the introduction, but there is no introduction to the interplay between the immune system and the production of NO.
In the context of cnidarian-dinoflagellate symbiosis, this section is even more important, as bleaching is proposed to be triggered by oxidative stress. A paragraph describing this interplay in the introduction would bridge the gap between the bioinformatic analysis and the laboratory essays, enhancing the text flow.
The use of DAF oxidation as a proxy for NO concentration could be explained in the introduction too.

Some phrases are misplaced and a general review of the manuscript could resolve this issues.
The following statements should be moved to the introduction:
Lines 255-260: 'We hypothesized that if a symbiont is co-opting host SRs to initiate
tolerogenic pathways (such as the TGFβ pathway) that dampen or prevent an immune response, blocking SR-ligand-binding capabilities could induce an immune response upon the addition of lipopolysaccharide (LPS). LPS is a MAMP that has been shown to induce an anemone immune response measured as increased nitric oxide (NO) production (Detournay et al. 2012; Perez & Weis 2006).'
Lines 313-314: 'Vertebrate SR-As are defined by a collagen domain coupled with most proteins containing either an SRCR domain or a CTLD at the C terminus (Bowdish & Gordon 2009)'
Lines 317-319: 'Human SR-Es are defined by the presence of only CTLDs (Zani et al. 2015). The human lectin-like oxidized low-density lipoprotein receptor 1 (LOX1) has an N-terminal cytoplasmic tail, a transmembrane domain and a single C-terminal CTLD (Canton et al. 2013)'
Lines 320-322: 'SR-Is in humans are defined by containing only SRCR domains in various numbers of repeats and are grouped into three classes: CD5, CD6 and CD163'

The following statements should be moved to the methods:
Line 373: '(to test for fucoidan toxicity to the animal)'
Lines 376-378: 'A second fucoidan experiment investigated the possible immune-regulation role of an SR in symbiosis maintenance. Symbiotic anemones were treated with increasing concentrations of fucoidan and were subsequently immune-challenged by incubation with LPS.'

Minor issues:
Lines 158/159: Either indicate it is Aipatsia pallida (as it was done throughout the manuscript) or remove the term 'species', for consistency. (I know this is not the authors' fault, anyway...)
Lines 162-166: There might be some confusion between experimental manipulation and bioinformatic manipulation. Change 'All resources were used without manipulation' for something like 'Published transcriptome/genome assemblies'.
Line 169: 'primarily from GenBank'. What's the other source?
Line 170: Most importantly, these sequences are queries for the Blast searches on cnidarian resources.
Line 178-179: not clear what the authors mean by 'Keyword searches using the terms SR, CD36, LMP2, SRCR, and scavenger were also performed.' Was it on the annotation provided by the original articles? How were then obtained? Was it different from the methodology presented?
Lines 190-191: Which databases?
Line 202. It's actually figure 2.
Line 260. Reinforce that 'Symbiotic anemones were incubated'
Line 286: 'a baseline colonization level for symbiotic anemones'. Shouldn't it be 'aposymbiotic anemones'?
Line 378-380: Change 'The FSW control-treated anemones had low levels of NO production, a proxy for an immune response, measured as MFI of the NO-specific probe DAF-FM DA in tentacles, in response to incubation in LPS.' to:
'The FSW control-treated anemones had low levels of NO production in response to incubation in LPS.'
Removed phrases should be in introduction/methods.

Experimental design

The manuscript clearly defines a meaningful research question, addressed in an original research.
Reporting of the methods can be improved with a clearer reporting of the bioinformatic analysis. blast and pfam searches should be explained separately, with respective queries and algorithm parameters (lines 169 – 197).
Description of the bioessays shall be clearer if the paragraph on immune response and NO production is included in the introduction.

The statistical model should be presented, with more explanation. It is hard to evaluate/understand what was done. Was it used random effects models for the individuals and polyps within individuals? It looks like a nested design with repeated measures is more appropriate for dealing with the lack of independence in lines 298-301.
As the test is based in differences between slopes, the slopes should be presented in the results (at least as supplementary data).

Validity of the findings

Results presented looks robust although the statistical methods needs a better description.
The only retention I have with the results published in this study is with the phylogenetic analysis of CD36 domain containing sequences. Results suggests that CD36-domain containing protein diversification occurred independently in cnidarians and bilaterians. Although this might be interesting, it contributes little to this manuscript. At least in the manner it is currently presented.
Furthermore, 'basal' metazoan sequences other than cnidarians are under-represented and the phylogeny is constructed with varying number of paralogous sequences. In this context, the fact that a ctenophore sequence groups together with sponges instead of cnidarians looks like a long-branch attraction artifact.
My recommendation is to remove it from the current manuscript and treat it in a separate study, where this analysis could be improved with a more comprehensive review of the available genomic resources and by the identification of orthologous sequences.

Additional comments

The manuscript succeeds in the task of identifying novel SR in cnidarians and suggesting a role for them in the cnidarian-dinoflagellate symbiosis and I'd like to compliment the authors for that.
Although the manuscript might be ‘self-contained' in it's present form, it would be interesting to have a larger discussion in the effects between the cnidarian immune system, the production of NO and the disruption of symbiosis under oxidative stress.
Would the oxidation burst in the dinoflagellate under heat stress causes the failure of the symbiont to regulate the host immune system?

It would be also interesting to evaluate the presence of the set of SR in non-cnidarian hosts. Riesgo et al (2014) presents transcriptomic resources for both symbiotic and aposymbiotic sponge Cliona varians. It would be interesting to include it in the bioinformatic analysis presented.


Riesgo, A., Peterson, K., Richardson, C., Heist, T., Strehlow, B., McCauley, M., Cotman, C., Hill, M. and Hill, A., 2014. Transcriptomic analysis of differential host gene expression upon uptake of symbionts: a case study with Symbiodinium and the major bioeroding sponge Cliona varians. BMC genomics, 15(1), p.1

---

## Round 0.2 · Minor Revisions

· Academic Editor

Minor Revisions

Your revised text contains the majority of the remarks of the referees. One remark is missing though. Please make to revise your text accordingly.

>20% identity in a protein domain does not mean a "well conserved among metazoans". On the contrary, it is a weak phylogenetic signal.

eg
Lines
375 Protein sequence alignments of the predicted SR-B-like proteins from
376 cnidarians, combined with a subset of vertebrate and invertebrate sequences, revealed that the
377 CD36 domain is highly conserved across metazoans. Cnidarian sequences showed moderate
378 homology to human SR-Bs, with 26-32%, 28-37% and 28-33% identity to human CD36, LMP2,
379 and SR-B1, respectively. Identities within the cnidarian group were substantially higher, ranging
380 from 39 to 95%, with the two Acropora species showing the highest homology to each other.
381 Cnidarian sequences showed between 21 and 27% identity to the predicted SR-B-like protein
382 sequence from the sponge, Suberites domuncula.

So please tone down this. For the rest is ready to be accepted. Sincerely. Fabiano

---

## Round 0.3 · accepted · Accept

· Academic Editor

Accept

Your article is now accepted for publication.